# Composition, Color Stability and Antioxidant Properties of Betalain-Based Extracts from Bracts of *Bougainvillea*

**DOI:** 10.3390/molecules27165120

**Published:** 2022-08-11

**Authors:** Qiang Wu, Xueying Fu, Zhuo Chen, Huafeng Wang, Jian Wang, Zhixin Zhu, Guopeng Zhu

**Affiliations:** 1Key Laboratory for Quality Regulation of Tropical Horticultural Crops of Hainan Province, College of Horticulture, Hainan University, Haikou 570228, China; 2Sanya Nanfan Research Institute of Hainan University, Hainan Yazhou Bay Seed Laboratory, Sanya 572025, China; 3Key Laboratory of Germplasm Resources Biology of Tropical Special Ornamental Plants of Hainan Province, College of Forestry, Hainan University, Haikou 570228, China

**Keywords:** *Bougainvillea*, colorants, antioxidant, betalains, betacyanins, betaxanthins, absorption spectra, stability, pH

## Abstract

Betalains in bracts of *Bougainvillea* are of great application potential as natural food colorants and antioxidants. This study explored the color, spectra, composition, storage stability, and antioxidant properties of betalain-based *Bougainvillea* bracts extracts (BBEs) to verify their application value. The results showed that *Bougainvillea* bract color variance is due to varied contents and proportions of betacyanins (Bc) and betaxanthins (Bx). Bc or Bx alone determined hues of purple or yellow, respectively; the co-existence of Bc and Bx would produce varied hues of red. BBEs showed bright color and good antioxidant properties under a wide pH range. The pH range of 5–6 was optimal for the highest color stability, and pHs 3–8 were optimal for stronger antioxidants. Bc mainly underwent color fading during storage, while Bx easily produced dark precipitates or melanism under strong acidic (pH < 4) or alkaline conditions (pH > 8). However, *Bougainvillea* Bx showed 3–4 times higher antioxidant ability than Bc. Different considerations for Bc and Bx are needed for varied application purposes. The purple bracts containing only Bc would be more suitable as colorant sources, while additional Bx can bring enhancement of antioxidant ability and richness of *Bougainvillea* extract color.

## 1. Introduction

Food colorants are widely used in industries of food, medicine and cosmetics [1,2]. Compared to the health concerns associated with synthetic pigments, natural plant pigments, including anthocyanins, betalains, and carotenoids, are all reported to have health values, largely due to their antioxidant abilities [3,4,5,6]. However, the instability of the natural plant pigments, including degradation and discoloration, greatly hinders their application [3,7,8].

Betalains, a kind of water-soluble plant pigments similar to anthocyanins, are mutually exclusive with anthocyanins in plants [9]. Betalains are much less reported due to the fact that betalainic plants are restricted to some plant families in the order Caryophyllales [9,10]. However, plants colored by betalains are common in daily life, such as red beet (*Beta vulgaris* L.), pitahaya (*Hylocereus polyrhizus*), *Amaranthus*, *Bougainvillea*, etc. Betalains are betalamic acid-derived alkaloids synthesized from tyrosine and are mainly categorized into two groups based on their conjugation partners: (1) betacyanins (Bc), conjugated by *cyclo*-dihydroxyphenylalanine (*cyclo*-DOPA) (Figure 1A), correspond to the red-hue color from red to violet; (2) betaxanthins (Bx), conjugated by amino acids or amines (Figure 1B), render a yellow-hue color from yellow to orange [10,11,12].

Oxidative stress is involved in the pathological processes of many conditions in the human body, such as tumors, cancer, arteriosclerosis, diabetes, and aging [10,11]. Betalains have attracted much attention due to their potent antioxidant capacity, which would provide chemo-prevention of oxidative stress [2,5,6,13]. At present, red beets are the main source of commercial betalains as food colorants [5,14]. However, the unfavorable earthy flavor from red beet preparations is unwelcome by many consumers [15,16]. Pitaya (*Hylocereus undatus*) is a good source of betalains, but it has too much pectin and is unsuitable under many circumstances. It is necessary to develop more alternatives from other species as sources of betalains. *Bougainvillea*, shrubs belonging to the Nyctaginaceae family, have become common ornamental plants in tropical and subtropical regions due to their easy cultivation, conspicuous bracts and long flowering period. Leaves of *Bougainvillea* were reported to have medicinal values such as antioxidative and anti-diabetes properties [17]. The bracts of *Bougainvillea* were also reported as anti-inflammatory [18]. The application values of *Bougainvillea* bracts as a potential betalain pigment source for natural antioxidants and colorants deserve further study.

Here, colorimeter and composition analyses were done for the bracts of representative *Bougainvillea* sp. with bright color. Then, the color, spectra, storage stability, and antioxidant properties were explored for *Bougainvillea* betalain-based extracts (**BBEs**) from three typical bract varieties at different pHs. From our results, the contents and proportions of Bc and Bx accounted for varied bract colors of *Bougainvillea*. The optimal pH range was determined for the storage of BBEs and also for maximal antioxidant ability. The considerations for using *Bougainvillea* bracts as potential sources of natural colorants and antioxidants were discussed.

## 2. Results

### 2.1. Colorimeter Analysis for the Bracts of Bougainvillea sp.

Six *Bougainvillea* sp. bearing seven representative colors were used (Figure 2A, details in Materials). The bracts were named after their color: “1 Deep Purple” and “2 Purple” have bright and distinct purple colors; “3 Mature Purple” and ”4 Young Red” were from two developmental stages of *B. spectabilis* Willd; while “5 Red”, “6 Pink Orange”, and “7 Orange” were from ornamental cultivars.

CIEL*a*b* parameters show the objective color evaluation for bracts of the *Bougainvillea* sp. (Figure 2B). From values of *a** and *b**, the bracts can be divided into three color sessions—the purple, orange, and red sessions. The purple session included “1 Deep Purple”, “2 Purple”, and “3 Mature Purple”, which have higher *a** values and lower b* values than the other two sessions. The orange session included “7 Orange”, which has the highest *b** and lowest *a** values. The red session included “4 Young Red”, “5 Red”, and “6 Pink Orange”. From values of *L**, “7 Orange” was brighter, and “5 Red” was darker than the others.

### 2.2. Spectral and Composition Analysis for Betalain-Based Extracts from Bougainvillea Bracts

From the spectral scanning of the BBEs, two main absorption peaks (λmax) appeared (Figure 3A). λmax around 540 nm (at 537 nm or 547 nm) shows the peak for Bc, and λmax around 480 nm shows the peak for Bx. Bracts from the purple session only showed peaks for Bc: “1 Deep Purple” had λmax at 547 nm, while “2 Purple” and “3 Mature Purple” had λmax at 537 nm. Bracts from the orange session (“7 Orange”) only showed the peak for Bx (480 nm). Bracts from the red session (“4 Young Red”, “5 Red”, and “6 Pink Orange”) all showed both peaks for Bc (537 nm) and for Bx (480 nm).

The composition analysis (Figure 3B) was done according to Kugler et al. [19]. The purple session only had Bc, the orange session only had Bx, and the red session possessed both betalains. However, the contents and the ratio of Bx and Bc (Bx/Bc) varied significantly for each *Bougainvillea* sp., similar to the concentrations of betalains in roots of red beet cultivars (200–2100 mg/kg FW) [20]. Notably, “5 Red” had the highest contents of betalains (2495 µg/g Bx and 1450 µg/g Bc, with a Bx/Bc ratio of 1.72). The Bx/Bc were similar for “4 Young Red” (1.38) and “6 Pink orange” (1.28), but the contents of “4 Young Red” (1112 µg/g Bx and 804 µg/g Bc) doubled that of “6 Pink Orange” (553 µg/g Bx and 432 µg/g Bc). Notably, the contents of Bc were comparable for “3 Mature Purple” (758 µg/g) and “4 Young Red” (804 µg/g), indicating the loss of Bx is the main reason for color change during the bract development.

### 2.3. Color and Spectra Analysis for Bougainvillea Betalain Extracts at Different pHs

The coloration of plant pigments is often severely affected by pH [21]. The BBEs from “3 Mature Purple”, “7 Orange”, and “4 Young Red” were chosen as typical samples from the purple, orange, and red sessions, respectively (Figure 4).

Here, “3 Mature Purple” represents a sample with the sole betalains of Bc. The BBEs gradually changed from magenta (pH 1–8) to brown (pH 9–11) and then to yellow (pH 12–13) (Figure 4A, Day 0). The corresponding absorption spectra were shown (Figure 4B). The spectra were almost overlapped from pH 1 to 6 (λmax = 537 nm), with a slight increase in the peak value from pH 1 to 3. Then, from pH 7 to 13, the absorbance within 380–460 nm gradually elevated, while the absorption peaks for Bc underwent gradual decline and bathochromic shift from pH 7 (λmax = 537 nm) to pH 12 (λmax = 551 nm), indicating structural transformation or degradation of Bc when pH > 7.

Here, “7 Orange” represents a sample with the sole betalains of Bx. The BBEs maintained a yellow hue (pH 1–13), but obvious brownish hues occurred around pH 9 (Figure 4C, Day 0). From the absorption spectra (Figure 4D), the curve underwent uplifts wholly from pH 1 to 3 and then became almost overlapped from pH 4 to 7. When pH > 8, the absorbance within 380–460 nm gradually elevated, while the peak at 480 nm declined slowly and eventually vanished at pH 12–13.

Here, “4 Young Red” has both Bc and Bx betalains. The BBEs gradually changed from red (pH 1–8) to brownish-red (pH 9–11) and then to yellow (pH 12–13) (Figure 4E, day 0), which is the additive color of “3 Mature Purple” and “7 Orange” under each pH. The corresponding absorption spectra (Figure 4F) also showed as the additive spectra for that of “3 Mature Purple” and “7 Orange”. For BBEs from pH 7 to 13, Bc peaks declined with gradual bathochromic shift (λmax from 537 nm to 551 nm), while no such λmax shift was shown for the Bx peaks (λmax = 480 nm).

### 2.4. Storage Stability for Bougainvillea Betalain Extracts at Different pHs

Storage stability for BBEs at different pHs was explored in the aspects of color (Figure 4) and absorption spectra (Appendix A). Thermal degradation kinetics, including the BBE degradation rate constant (k) and half-life time (t_1/2_), were also calculated based on the peak values of daily collected spectra (Table 1). Bc and Bx showed different degradation properties during storage at room temperature.

For “3 Mature Purple”, the magenta BBEs quickly turned yellow within one day when pH > 8, while the color gradually faded when pH < 8, and the color was most stable for BBEs under pH 3–6 (Figure 4A, Day 20). Thermal degradation parameters followed the above results, with the longest t_1/2_ for BBEs under pH 3–6, especially at pH 4 (t_1/2_ = 13.48 days).

For “7 Orange”, only BBEs at pH 5–7 maintained acceptable yellow color. When pH < 5, the yellow color decayed and became colorless, accompanied by the formation of some dark precipitates. When pH > 7, obvious melanism occurred for the solutions, especially at pH 8–9 (Figure 4C).

The BBEs of “4 Young Red” showed very clear additive properties for that of “3 Mature Purple” and “7 Orange”. The mixture of the Bc (magenta) and Bx (yellow) produced the red color (Bc + Bx). At pH 1 of Day 2, the BBE was magenta when the Bx peak vanished while the Bc peak was retained (Figure 4E, Day 2). When pH > 7, melanism also occurred, just as those for “7 Orange”.

The optimal pH range for higher color stability was easily determined on the aspect of color change. The BBEs were most stable at pH 3–6 for “3 Mature Purple” (Figure 4A), pH 5–7 for “7 Orange” (Figure 4C), and pH 3–7 for “4 Young Red” (Figure 4E). The pH 5–6 were the shared optimal range for BBE color stability for all three *Bougainvillea* bract varieties.

However, on aspects of absorption spectra (Appendix A), the results became complicated. When at pH 1–3, the degradation of Bc or Bx peaks followed the 1st-order reaction kinetics. When pH > 8, the Bc or Bx peaks vanished within one day’s time, and melanism occurred for Bx. When at pH 3–8, the curve change becomes confusing: structural change, maybe polymerization or condensation, prevented the decline of curves for Bc and caused the uplifts of curves for Bx. For the thermal degradation parameters of k and t_1/2_, only Bc peaks at pH 1–8 and Bx peaks at pH 1–3 can be calculated (Table 1).

### 2.5. Antioxidant Activities for Bougainvillea Betalain Extracts at Different pHs

Potent antioxidant capacity for betalains has been reported in cactus pear, red beets, etc. [4,22]. Here, the ABTS free radical scavenging abilities were assessed for BBEs of “3 Mature purple”, “4 Young Red”, and “7 Orange” at different pHs (Figure 5).

For each *Bougainvillea* bract variety, multiple concentrations of bract extracts were used, and the concentration with maximal radical scavenging between 70 and 90% was shown (Figure 5). From the concentration difference, Bx was indicated to have higher antioxidant capacity than Bc (about 3–4 times higher), given that our composition analysis showed similar Bc contents in “3 Mature Purple” (758 µg/g) and “4 Young Red” (804 µg/g), and comparable Bx contents in “4 Young Red” (1112 µg/g) and “7 Orange” (1166 µg/g). The antioxidant ability was strongest for bracts of “4 Young Red” (working concentration 16.6 mg/mL), which had a joint presence of Bc and Bx. The lowest antioxidant ability was shown in “3 Mature Purple”(working concentration 100 mg/mL), which contained solely Bc.

BBEs rendered considerable antioxidant ability under a wide pH range of pH 1–13 (>35%), with pH 3–8 being the optimal range for the highest scavenging percentage (75–90%). BBEs showed slightly lower scavenging percentages at pH 1–2 (65–75%). When pH > 8, the antioxidant ability declined with the rise of pH. However, a considerable scavenging percentage (35–52%) remained even for BBEs at pH 13, which was impossible for anthocyanins [21].

## 3. Discussion

### 3.1. How to Distinguish Betalains from Anthocyanins

Although the biosynthesis and chemical structures are completely different for betalains and anthocyanins, they share many similarities and have been suggested to be functional homologs [23,24]. They are both water-soluble, synthesized in the cytoplasm, and stored as glycosides in vacuoles; they have comparable absorption spectra and colors, similar histological locations in plant tissues, and even similar functions for plants such as coloring, antioxidation, and photo-protection [11,23,24].

Betalains are often misreported as anthocyanins in daily life. The confusion is mainly between the purplish Bc and red-hue anthocyanins, which have similar absorption spectra, and hence similar colors. How to distinguish betalains from anthocyanins? The following tips can be used as guidance.

First, betalainic plants are restricted to the order of Caryophyllales, and betalains can be regarded as significant chemo-taxonomical indexes [24]. Notably, even in Caryophyllales, two families (the Caryophyllaceae and Molluginaceae) contain members that produce anthocyanins instead of betalains, *Lychnis* and *Dianthus* (e.g., lychee, carnations, etc.) [9,23]. Although it is reported that dysfunction of ANS accounts for the absence of anthocyanins in betalainic plants, the fact of mutual exclusion for betalains and anthocyanins in plants is still a mysterious puzzle that needs to be clarified [11].

Second, betalains and anthocyanins each have their distinct color range. A blue color can be conferred by anthocyanins but not by betalains. Meanwhile, the yellow color exemplified by Bx betalains is rare for anthocyanins under natural vacuolar pH [21,25]. It is also easy to distinguish the yellow-colored Bx from carotenoids, as Bx are water-soluble, while carotenoids are lipophilic.

Third, the color stability under various pHs is a handy indicator of the difference between the two pigments. Unlike betalains and carotenoids, anthocyanins tend to change color depending on their physical environment and turn colorless, green-blue, or yellow under neutral or alkaloid pHs [21,25]. When adding a drop of a weak alkaloid (such as baking soda) to the pigments, anthocyanins turn colorless or blue-green, while betalains do not change color. The high color stability is also demonstrated by one common example—urine and feces often turn red after we consume purple pitaya, but not after intake of anthocyanin-rich plants.

### 3.2. Contents and Proportion of Bc and Bx Betalains Account for the Varied Colors of Bougainvillea

There are about 80 well-characterized plant-derived betalains, including Bc and Bx [12]. Betalain patterns of differently colored *Bougainvillea* sp. were studied by HPLC–DAD–ESI–MS [19,26]. The major *Bougainvillea* Bx was reported as Dopa-Bx, followed by small amounts of histidine-Bx and putrescine-Bx [19]. At least 30 types of Bc were reported for *Bougainvillea,* mainly the derivatives of betanidin 6-*O*-glucoside [19,26].

*Bougainvillea* are ornamental plants well-known for their colorful bracts. The broad color palette of *Bougainvillea* bracts includes white, pink, red, magenta, yellow, orange, etc. From our results, contents and proportions of Bc and Bx betalains account for the above-varied colors: the presence of Bc solely accounts for varied purple hues; Bx solely renders hues of yellow-orange; while the joint presence of Bc and Bx produces varied hues of red depending on their contents and proportions. Similar results were also drawn for other betalainic plants, such as *Gomphrena* petals [19] and cactus juices [27]. Loss of Bx accounts for the developmental color change from “Young Red” to “Mature Purple”. Combinations of Bx and Bc in aspects of contents and proportions can bring a great variation of bract color in the breeding of ornamental *Bougainvillea* sp.

### 3.3. Bougainvillea Betalains as Potential Natural Colorants and Antioxidants

Betalains from red beets and pitaya are reported as natural colorants with multiple beneficial health effects [2,5,12,13,27]. From our assay of ABTS free radical scavenging, BBEs also showed good antioxidant properties in a wide pH range (pHs 1–13), and pHs 3–8 were optimal for stronger antioxidants.

From the aspect of color, our results showed *Bougainvillea* betalain extracts could display a broad color range by combinations of Bc and Bx—the purple session with solely Bc (storage stable at pH 3–6), the orange session with solely Bx (stable at pH 5–7), and the red session with both Bc and Bx (stable at pH 3–7). The range of pH 5–6 is the safest storage condition for high color stability for all *Bougainvillea* betalain sessions. Our above optimal pH results for maximum stability are consistent with studies in other betalainic plants, such as red beet and cactus juices [7,16,27]. In contrast, anthocyanins are usually most stable at pH < 3 [25]; this would be an advantage for betalains when foods of low acidity are to be stained.

However, one must be cautious regarding the occurrence of melanism for *Bougainvillea* Bx. In our assay of storage stability, BBEs containing Bx easily produced dark precipitates or melanin under strong acidic (pH < 4) or alkaline conditions (pH > 8). Dopa-Bx is the dominating Bx for *Bougainvillea* bracts [19]. The melanization pathway between L-tyrosine and dopachrome was reported [28]. The degradation of Dopa-Bx into dopachrome and L-tyrosine would easily lead to the generation of melanin in vitro. Quick metabolism of dopachrome may account for the lasting bright color of *Bougainvillea* bracts in planta irrespective of the betalain profile [11,12]. However, when extracted for application, special caution for melanism should be paid to *Bougainvillea* varieties containing Bx. The recommended pH 5–6 storage condition could be applied.

To exclude problems of melanism, the purple *Bougainvillea* varieties containing only Bc would be more suitable as colorant sources, as Bc mainly underwent color fading during storage. However, *Bougainvillea* Bx showed 3–4 times higher radical scavenging ability than *Bougainvillea* Bc. Additional Bx enhances the antioxidant ability and richness of *Bougainvillea* bract extract color.

## 4. Materials and Methods

### 4.1. Materials

Six *Bougainvillea* sp. bearing seven representative colors were used. The subspecies were identified as “1 Deep Purple” (*B. glabra* Choisy); “2 Purple” (*B.* cv. Sanderiana); “3 Mature Purple” and “4 Young Red” (*B. spectabilis* Willd); “5 Red” (*B. spectabilis* Flame); “6 Pink Orange” (*B.* cv. Auratus); and “7 Orange” (*B. × buttiana* Golden Glow). Among the above six subspecies, the bracts of *B. spectabilis* Willd were red when young (“4 Young Red”) and gradually turned purple when mature (“3 Mature Purple”), which is similar to the “juvenile red fading” phenomenon in leaves of many trees [29]. Here, “1 Deep Purple”, “2 Purple”, “3 Mature Purple”, and “4 Young Red” were collected on the campus of Hainan University. Furthermore, “5 Red”, “6 Pink Orange”, and “7 Orange” were kindly provided by Dr. Zhou Yang from his collections. Fresh bracts were collected, and the central veins were removed for subsequent pigment extraction to exclude occasional chlorophylls on the veins.

### 4.2. Colorimeter Analysis

For an objective color evaluation, uniform CIELab color space parameters were measured for each *Bougainvillea* variety with a hand spectrophotometer (LS173B, Linshang Technology, Co., Ltd., Shenzhen, China). Values of *L**, *a**, and *b** were recorded. *L** represents lightness with values from black (0) to white (100); *a** represents greenish (−) and reddish (+) hues as the value increases from negative to positive; and values of *b** represent bluish (−) and yellowish (+) hues. Three replicates from different bracts were recorded for each *Bougainvillea* variety.

### 4.3. Spectral and Composition Analysis for Bougainvillea Betalain-Based Extracts

Betalains are water-soluble pigments and were better extracted by water than by 80% ethanol or citrate-phosphate buffer (pH 5.5) [16]. Here, bracts of the *Bougainvillea* varieties were extracted with sterilized ddH_2_O as recommended [16]. Three pieces of bracts from each *Bougainvillea* variety were collected. After removal of the central veins, the remaining bracts were weighed (0.1–0.3 g), ground with 6 mL ddH_2_O, and then filtered. As betalains were reported to have a λmax around 540 nm (for Bc) or 480 nm (for Bx) [10], A_540_ or A_480_ were measured for the supernatants with ddH_2_O as a blank (UV-2100, UNICO). Appropriate dilutions were applied to the supernatants so that the final A_540_ or A_480_ values were around 1. The derived samples were subjected to continuous spectral scanning from 380 to 700 nm (UV-2600, SHIMDZU), and a more accurate λmax was determined for extracts from each *Bougainvillea* variety.

From the derived spectra, absorption peaks at 480 nm represent the existence of Bx, and peaks at 537 nm or 547 nm show the existence of Bc. Contents of betalains were calculated with Equations (1) and (2) when the peaks appeared as reported [4,19].
Bc contents (μg/gFW) = (A_480_ × DR × MW × V × 1000)/(ε × L × m)(1)
Bx contents (units/gFW) = (A_537/547_ × DR × MW × V × 1000)/(ε × L × m)(2)

In the above formulas, DR is the dilution rate; MW is the molecular weight (550 g mol^−1^ for betanin and 390 g mol^−1^ for Dopa-Bx); V is the extraction volume (mL); ε is the molar extinction coefficient (60,000 L mol^−1^·cm^−1^ for betanin and 48,000 L mol^−1^·cm^−1^ for Dopa-Bx); L is the path length (1 cm) of the cuvette; and “m” is the fresh weight (FW). The means and standard error (SE) for data of three biological replicates were presented.

### 4.4. Color Stability Analysis for Bougainvillea Betalain-Based Extracts at Different pHs

About 5 g *Bougainvillea* bracts were ground with 15 mL ddH_2_O. The homogenized mixture was filtered by squeezing through sterilized gauze. After further centrifugation at 10,000 rpm for 5 min, the supernatants were collected as stock solutions that contained high betalain contents. Then, the dilution rate (DR, from 20 to 40) was determined for each stock solution to obtain final values of A_540_ or A_480_ around 1.

The above stock solutions were diluted by buffers at different pHs by DR to prepare BBEs with A_540_ or A_480_ around 1. Citrate-phosphate buffer (pH 2.6, 3.0, 4.0, 5.0, 6.0, and 7.0) were prepared by varied ratios of 0.1 M citric acid and 0.2 M Na_2_HPO_4_ solutions. Citrate-phosphate-HCl buffer (pH 1.0 and 2.0) was prepared by titrating the pH 2.6 buffer with HCl to lower pHs. Phosphate buffer (pH 8.0) was prepared by a mixture of 0.2 M Na_2_HPO_4_ and 0.2 M Na_2_HPO_4_ solutions. Glycine-NaOH buffer (pH 9.0, 10.0, 11.0, 12.0, and 13.0) was prepared by varied ratios of 0.2 M glycine and 0.2 M NaOH solutions

According to former reports [13,16] and our pre-experiments (data not shown), light and high temperature accelerate pigment degradation and color change, while low temperatures of 4 °C greatly prolong the half-life time (t_1/2_) of color decay to more than a year. The derived BBEs were placed in the dark at room temperature (23–27 °C), conditions that would allow a comparison of the effect of pH within one month. For observation of color decay during storage, aliquots of 2 mL solution were taken into fresh tubes and arranged to take pictures. Another set of 200 μL aliquots was added to the microplate wells and subjected to continuous spectral scanning from 380 to 700 nm (Infinite 200Pro, Tecan i-control) from day 0 to day 15. The assays were repeated three times, and measurements were performed in triplicate.

### 4.5. Thermal Degradation Kinetics

The BBE degradation rate constant (k) and half-life time (t_1/2_) were calculated based on the peak values for Bx and Bc of daily collected spectra. OD values were recorded at the absorption peaks (λmax around 480 nm or 537 nm). The thermal degradation kinetics were analyzed as previously reported [21]. The rate constant of BBE degradation (k) and half-life time (t_1/2_) were calculated with Equations (3) and (4).
ln(A_t_/A_0_) = −k × t,(3)
t_1/2_ = −ln0.5 × k^−1^,(4)

In the equations, A_t_ is the absorbance at time t (day); A_0_ is the initial absorbance.

### 4.6. Measurement of Antioxidant Activities for BBEs at Different pHs

BBEs at different pHs were assayed for antioxidant activities by methods of DPPH (2,2′-diphenyl-1-picrylhydrazyl) and ABTS (2,2′-azino-bis (3-ethylben-zothiazoline-6- sulphonicacid)) radical scavenging as previously reported [8] with some modifications.

For DPPH radical scavenging abilities, DPPH working solutions (1 × 10^−4^ mol/L) were prepared on the day of the assay by dissolving 3.9432 mg DPPH in 100 mL absolute ethanol assisted by ultrasonication. Then, a 10 μL sample was mixed with 1990 μL DPPH working solution and placed in the dark at room temperature. A_517_ were collected after 30 min.

For ABTS radical scavenging abilities, ABTS stock solution was prepared by mixing ABTS (7 mmol/L) with potassium persulfate (2.45 mmol/L). The mixture was left in the dark for 12–16 h to form a dark green stock solution with free radicals. The stock solution was then diluted with ddH_2_O to OD_734_ at 0.7 ± 0.05 before usage. Then, a 10 μL sample was mixed with 1990 μL of ABTS working solution. The mixed solutions were placed in the dark at room temperature. A_734_ were collected after 30 min.
Percentage of DPPH/ABTS radical scavenging (%) = [(A_blank_ − A_sample_)/A_blank_] × 100%,(5)

A_blank_ represents the absorption of the control sample, A_sample_ represents values for the tested sample. The results of the DPPH scavenging assays were discarded as the mixture would turn purple/orange when antioxidants occurred for BBEs with pH > 5, which interfered with the value of A_517_. In the ABTS scavenging assays, obvious fading of the green color was shown for antioxidants. The means and SE for the data of three independent replicates in the ABTS scavenging assays were presented.

## 5. Conclusions

From our results, the richness of *Bougainvillea* bract color results from varied existence and content proportion of Bc and Bx betalains. The exclusive existence of Bc or Bx determined the hues of purple or yellow, respectively. The co-existence of Bc and Bx produced varied hues of red. The results provide insights into the breeding of ornamental *Bougainvillea* varieties.

BBEs showed bright color and good antioxidant properties under a wide pH range. The pH 5–6 was optimal for the highest color stability, and pHs 3–8 were optimal for stronger antioxidants. Bc and Bx showed different degradation properties during storage at room temperature. Bc mainly underwent color fading during storage, while Bx easily produces dark precipitates or melanism under strong acidic (pH < 4) or alkaline conditions (pH > 8). However, *Bougainvillea* Bx showed 3–4 times higher radical scavenging ability than Bc. Different considerations for Bc and Bx are needed for varied application purposes. The purple bracts containing only Bc would be more suitable as colorant sources. In contrast, additional Bx can enhance the antioxidant ability and color richness. The results of this study provide guidance for the application of *Bougainvillea* betalains as natural colorants and antioxidants in the food, pharmaceutical, and cosmetics industries.

## Figures and Tables

**Figure 1 molecules-27-05120-f001:**
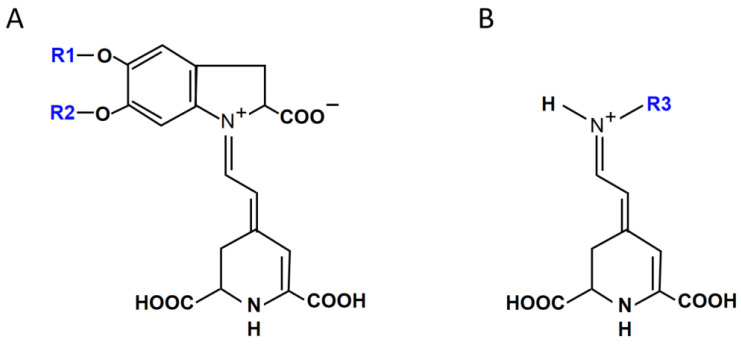
Structure of betacyanins (Bc) (**A**) and betaxanthins (Bx) (**B**). R1 and R2: hydrogen or sugar moieties; R3: amino acid or amine group. After [9,12], modified.

**Figure 2 molecules-27-05120-f002:**
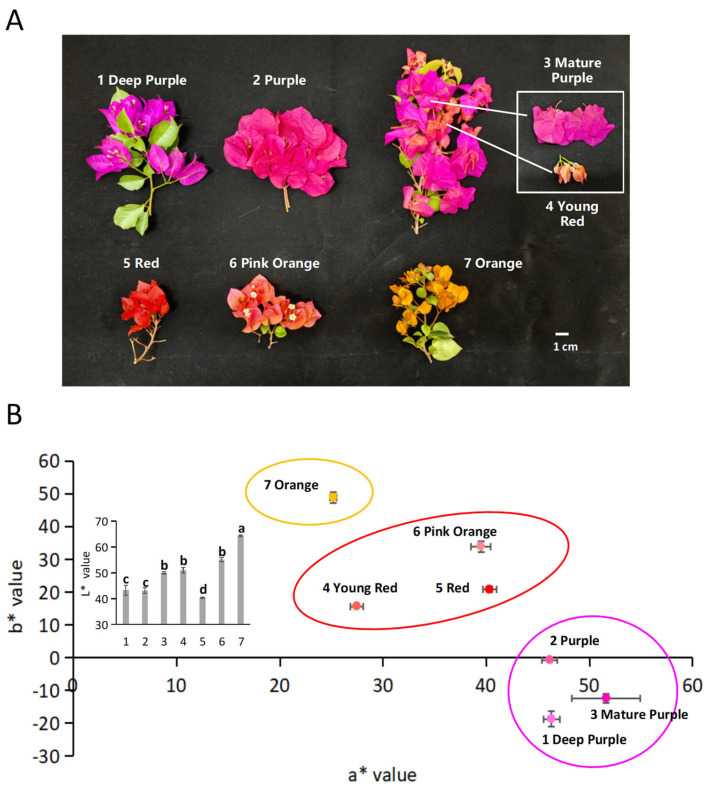
Colorimeter analysis for the bracts of *Bougainvillea* sp. (**A**) Six *Bougainvillea* sp. bearing seven representative colors. (**B**) CIEL*a*b* parameters for the bracts of *Bougainvillea* varieties. The inset histogram shows the *L** values. All values were based on three independent replicates. The difference significances for *L** were done by one-way ANOVA and Duncan’s test at *p* < 0.05.

**Figure 3 molecules-27-05120-f003:**
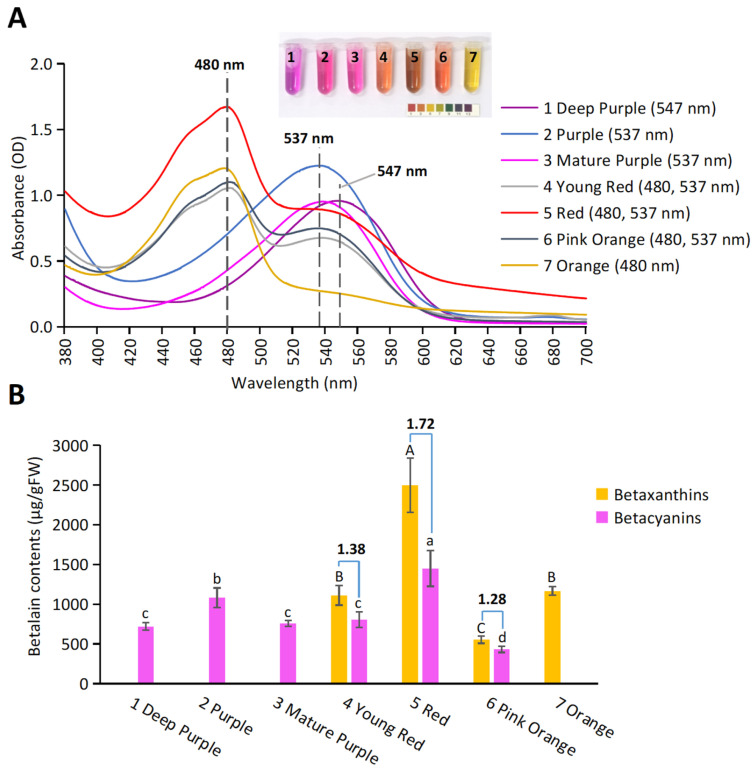
Absorption spectra scanning (**A**) and the betalain composition analysis (**B**) for the BBEs from *Bougainvillea* bracts. For (**A**), the tubes show the color of the scanned BBEs and the wavelength value in the bracket indicates the λmax. The universal pH indicator paper was shown as a color reference. For (**B**), ratios of Bx/Bc were shown for bracts with both Bx and Bc. The values were based on three independent replicates. Significant differences were calculated by one-way ANOVA and Duncan’s test at *p* < 0.05. The uppercase and lowercase letters show the significant differences among samples for Bx and Bc, respectively.

**Figure 4 molecules-27-05120-f004:**
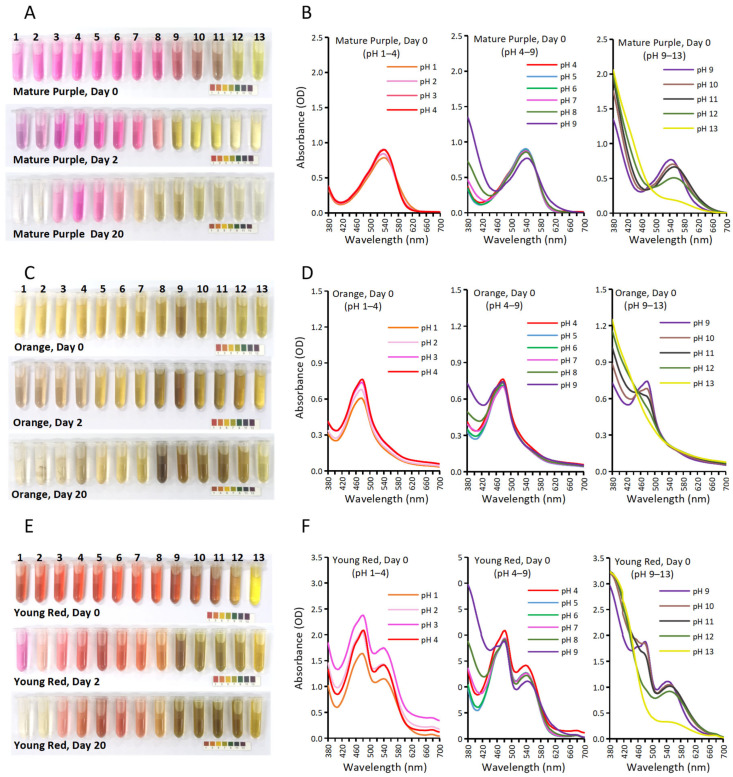
Color stability and spectra analysis for BBEs at different pHs. Colors of BBEs at different pHs on storage on Day 0, Day 2, and Day 20 were shown for “3 Mature Purple” (**A**), “7 Orange” (**C**), and “4 Young Red” (**E**). Their respective spectra (**B**,**D**,**F**) on Day 0 were demonstrated. The universal pH indicator papers were shown as color references.

**Figure 5 molecules-27-05120-f005:**
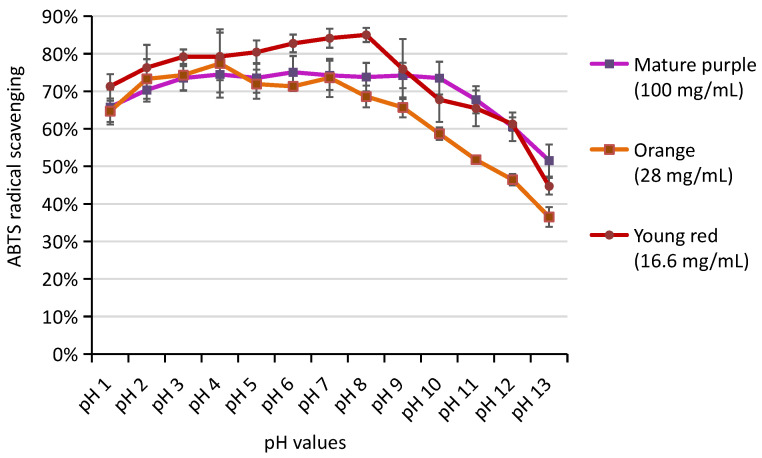
ABTS radical scavenging percentage for BBEs at different pHs. The concentrations of the bract extracts were shown in the brackets for each material.

**Table 1 molecules-27-05120-t001:** Kinetic parameters of thermal degradation for BBEs at different pHs.

pH Values	Betacyanin (Mature-Purple)		Betacyanin (Young-Red)		Betaxanthin (Young-Red)		Betaxanthin (Orange)	
	k (d−1)	t1/2 (day)	k (d−1)	t1/2 (day)	k (d−1)	t1/2 (day)	k (d−1)	t1/2 (day)
pH 1	0.209	3.44	0.213	3.39	0.651	1.12	0.098	8.93
pH 2	0.283	2.78	0.337	2.54	0.411	2.35	0.113	7.72
pH 3	0.059	13.14	0.108	17.45	0.411	2.90	0.096	8.65
pH 4	0.055	13.48	<0	–	<0	–	<0	–
pH 5	0.057	12.68	<0	–	<0	–	<0	–
pH 6	0.060	11.67	<0	–	<0	–	<0	–
pH 7	0.105	6.67	<0	–	<0	–	<0	–
pH 8	0.154	4.69	<0	–	<0	–	<0	–
pH 9	–	<1	<0	–	<0	–	<0	–
pH 10	–	<1	–	<1	–	<1	–	<1
pH 11	–	<1	–	<1	–	<1	–	0
pH 12	–	0	–	0	–	0	–	0
pH 13	–	0	–	0	–	0	–	0

“–” means values unable to be determined. The half-life time (t_1/2_) was shown as t_1/2_ = 0 for those peaks that vanished instantly for Bx or Bc, and as t_1/2_ < 1 for those with peaks that vanished within one day. For those with the peak values increased during storage, k < 0 and t_1/2_ could not be determined.

## Data Availability

Not applicable.

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
