# Peer review of "Composition, Color Stability and Antioxidant Properties of Betalain-Based Extracts from Bracts of Bougainvillea"

_molecules, 2022, doi:10.3390/molecules27165120_

Round 1

Reviewer 1 Report

I have reviewed the manuscript entitled " Composition, Color Stability and Antioxidant Properties of Betalain-based Extracts from Bracts of Bougainvillea", by Wu et al. This work is well presented and easy to read. Experiments are well planned and the analyses were affected by appropriate methods. There is sufficient discussion of the results obtained. I recommended the publication of this work in Molecules after minor revisions.

Detailed remarks about the text are as follows:

Page 9 line 307 Extraction processes: How were the extraction conditions set? If it was made according to any reference, it should be stated. How are the variables (time, temperature, etc.) determined?

4.4 Color Stability Analysis for Bougainvillea Betalain-based Extracts at different pHs: How was the Color Stability analysis set? How conditions were determined? Please indicate references; otherwise please discuss how these conditions were chosen, as this is very important for the results.

Most importantly, there are no data on the number of repetitions.

Give correlation data between antioxidant activities and BBEs at different pHs

Statistical analysis should be included in the study. There is no information about statistical analysis.

Conclusions should be more concise, with some considerations, for example, about the importance of the result, what they mean for the industry, which is the most significant novelties of the results.

Author Response

Dear reviewer, Thank you for your comments. We revised the manuscript as suggested. Point 1: Page 9 line 307 Extraction processes: How were the extraction conditions set? If it was made according to any reference, it should be stated. How are the variables (time, temperature, etc.) determined? Response 1: Thank you for your question. The extraction conditions set were set as reference [16] (Castellar et al., 2003), which was stated (Line 318-319) and revised (Line 320). In reference [16], multiple extraction conditions were compared and extraction with pure ddH2O showed the best efficiency. With adequate grinding, betalains quickly dissolved into water and the plant residues turned pale. The bracts were ground in room temperature and filtered immediately, which can be easily applicable for daily life or large scale extraction. Point 2: 4.4 Color Stability Analysis for Bougainvillea Betalain-based Extracts at different pHs: How was the Color Stability analysis set? How conditions were determined? Please indicate references; otherwise please discuss how these conditions were chosen, as this is very important for the results. Most importantly, there are no data on the number of repetitions. Response 2: Revised as suggested. The Color Stability analysis were stated in Line 355-358, by two methods (highlighted by red; 1. pictures were taken every day for color change observation; 2. absorption spectra were scanned every day). Determination of storage conditions were added (Line 350-355). The storage condition (in the dark at room temperature) were easily applicable, and were chosen for our assays. We repeated the assay several times and the number of repetitions were shown at the end (Line 359). Point 3: Give correlation data between antioxidant activities and BBEs at different pHs Response 3: The antioxidant activities were shown as “Percentage of ABTS radical scavenging (%)”, which is good presentation of correlation between antioxidant activities and BBEs at different pHs. Point 4: Statistical analysis should be included in the study. There is no information about statistical analysis. Response 4: Revised as suggested. Statistical analysis and repetition information were stated at the end for each section of “4. Materials and Methods.”(highlighted as red; Line 315, 336, 359, 386-387). The statistical analysis for difference significances were added at the end of the figure legends. (Figure 2, Line 94; Figure 3, Line 119-120) Point 5: Conclusions should be more concise, with some considerations, for example, about the importance of the result, what they mean for the industry, which is the most significant novelties of the results. Response 5: Revised as suggested.

Reviewer 2 Report

This manuscript describes the composition, color stability, and antioxidant properties of bracts of Bougainvillea. The described findings are interesting and have scientific merit to be published in Molecules journal. However, there are several concerns about the data presentation, description, and interpretation, as follows. I suggest adding the structures of betacyanins (Bc) and betaxanthins (Bx) to the manuscript. how do you know that the hues of red and yellow come exclusively from Bc and Bx and not from other pigments such as flavonoids or carotenoids? (as described in the manuscript "represents a sample with the sole existence of Bc betalains"). How to distinguish betalains from anthocyanins using spectroscopy/spectrometry techniques?

Author Response

Dear reviewer,

Thank you for your comments. We revised the manuscript as suggested.

Point 1: I suggest adding the structures of betacyanins (Bc) and betaxanthins (Bx) to the manuscript.

Response 1: Thank you for your suggestion. We added Figure 1 as suggested.

Point 2: How do you know that the hues of red and yellow come exclusively from Bc and Bx and not from other pigments such as flavonoids or carotenoids? (as described in the manuscript "represents a sample with the sole existence of Bc betalains").

Response 2: Thank you for your question. Here, the “sole” were mainly used to distinguish samples from those with coexistence of Bc and Bx. “The sole existence of Bc/Bx betalains” were revised as “the sole betalains of Bc/Bx” (Line 126, 134, and 146).

Also, for your questions of other possible pigments.

First, for flavonoids, the existence of anthocyanins can be excluded. We added in “1. Introduction” to emphasize the exclusive existence of betalains and anthocyanins (Line 39). Other flavonoids (not including anthocyanins) can not be excluded from the aqueous extracts of BBEs, but they are usually colorless or of very pale yellow color and are not really deem as pigments.

Second, for chlorophylls and carotenoids. Some Bougainvillea bract varieties are green or have green veins, and do have chlorophylls and carotenoids, just as green leaves. In our assays, none of the bract varieties are green and all the the central veins were removed for subsequent pigment extraction to exclude occasional chlorophylls on the veins (Line 307-308). Also, both chlorophylls and carotenoids are lipophilic. The BBEs were extracted with water, the absence of chlorophylls or carotenoids can be verified from the absence of their characteristic absorption peaks shown by the absorption spectra.

Point 3:How to distinguish betalains from anthocyanins using spectroscopy/spectrometry techniques?

Response 3: As stated in Line 223-225 (highlighted as red), the confusion is mainly between the purplish Bc and red-hue anthocyanins, which have similar absorption spectra. They are difficult to be distinguished by sole spectroscopy/spectrometry techniques. However, with the extra three tips provided in “3.1 How to distinguish betalains from anthocyanins”, the two kinds of pigments can be distinguished. Especially, as stated in Tip 3 (Line 242-243), by adding a drop of the weak alkaloid, anthocyanins would change color, while betalains would not.